# Photoperiod Induces the Epigenetic Change of the *GNAQ* Gene in OVX+E_2_ Ewes

**DOI:** 10.3390/ijms242216442

**Published:** 2023-11-17

**Authors:** Wei Wang, Xiaolong Du, Mingxing Chu, Xiaoyun He

**Affiliations:** State Key Laboratory of Animal Biotech Breeding, Institute of Animal Science, Chinese Academy of Agricultural Sciences (CAAS), Beijing 100193, China; wangw8182@163.com (W.W.); dududu1201666@gmail.com (X.D.)

**Keywords:** sheep, photoperiod, *GNAQ*, DNA methylation, histone acetylation

## Abstract

GNAQ, a member of the alpha subunit encoding the q-like G protein, is a critical gene in cell signaling, and multiple studies have shown that upregulation of *GNAQ* gene expression ultimately inhibits the proliferation of gonadotropin-releasing hormone (GnRH) neurons and GnRH secretion, and ultimately affects mammalian reproduction. Photoperiod is a key inducer which plays an important role in gene expression regulation by affecting epigenetic modification. However, fewer studies have confirmed how photoperiod induces epigenetic modifications of the *GNAQ* gene. In this study, we examined the expression and epigenetic changes of GNAQ in the hypothalamus in ovariectomized and estradiol-treated (OVX+E_2_) sheep under three photoperiod treatments (short photoperiod treatment for 42 days, SP42; long photoperiod treatment for 42 days, LP42; 42 days of short photoperiod followed by 42 days of long photoperiod, SP-LP42). The results showed that the expression of GNAQ was significantly higher in SP-LP42 than in SP42 and LP42 (*p* < 0.05). Whole genome methylation sequencing (WGBS) results showed that there are multiple differentially methylated regions (DMRs) and loci between different groups of GNAQ. Among them, the DNA methylation level of DMRs at the CpG1 locus in SP42 was significantly higher than that of SP-LP42 (*p* < 0.01). Subsequently, we confirmed that the core promoter region of the *GNAQ* gene was located with 1100 to 1500 bp upstream, and the DNA methylation level of all eight CpG sites in SP42 was significantly higher than those in LP42 (*p* < 0.01), and significantly higher than those in SP-LP42 (*p* < 0.01), except site 2 and site 4 in the first sequencing fragment (*p* < 0.05) in the core promoter region. The expression of acetylated *GNAQ* histone H3 was significantly higher than that of the control group under three different photoperiods (*p* < 0.01); the acetylation level of sheep hypothalamic GNAQ genomic protein H3 was significantly lower under SP42 than under SP-LP42 (*p* < 0.05). This suggests that acetylated histone H3 binds to the core promoter region of the *GNAQ* gene, implying that GNAQ is epigenetically regulated by photoperiod through histone acetylation. In summary, the results suggest that photoperiod can induce DNA methylation in the core promoter region and histone acetylation in the promoter region of the *GNAQ* gene, and hypothesize that the two may be key factors in regulating the differential expression of GNAQ under different photoperiods, thus regulating the hypothalamus–pituitary–gonadal axis (HPGA) through the seasonal estrus in sheep. The results of this study will provide some new information to understand the function of epigenetic modifications in reproduction in sheep.

## 1. Introduction

Photoperiod is thought to be the main factor controlling the onset of sexual intercourse in animals with seasonal breeding patterns in temperate regions [1,2]. For most vertebrates living outside the tropics, seasonality is determined by tracking the annual cycle (day length) of photoperiodic variation. Depending on whether the day length is above or below a critical threshold, individuals can accurately determine the time of year and induce appropriate seasonal adaptations [3,4]. Light information is transmitted by the monosynaptic retinohypothalamic tract (RHT) to the master circadian clock located in the suprachiasmatic nucleus (SCN) of the hypothalamus [5,6,7]. The SCN transmits temporal information to secondary oscillators in the brain, such as the pineal and pituitary glands, and these brain regions transmit temporal information to other parts of the body [4]; this ultimately causes a pulsatile release of GnRH in the hypothalamus to regulate seasonal estrus activity. In a previous study on estrous regulation in sheep, our group found that the expression of the GNAQ (guanine nucleotide-binding protein q polypeptide) gene was significantly increased in the hypothalamus of sheep under short daylight conditions [8]. GNAQ is a member of the α subunit encoding of the q-class G protein and is a key gene vital in cell signaling. Previous studies have shown that G protein-coupled receptor GPR54 can regulate GnRH secretion by activating downstream signaling molecules via the Gαq (functional units of Gq proteins encoded by GNAQ) signaling pathway [9,10,11]. Wettschureck et al. [12] found via studying prepared Gαq knockout mice that mice with deletions of the *GNAQ* and *GNA11* genes died shortly after birth, despite being morphologically normal mice. In these mice, the proliferation of pituitary growth hormone cells was strongly impaired, and levels of hypothalamic growth hormone-releasing hormone (GHRH), an important stimulator of growth hormone cell proliferation, were also significantly reduced. Normal proliferation could be restored only by increasing exogenous GHRH, demonstrating that the Gq/11 signaling pathway is required for normal hypothalamic function. This suggests that *GNAQ* may influence the reproductive hormones produced by the hypothalamus and thus regulate physiological changes in animal reproduction. This was also demonstrated in a follow-up study, in which researchers found that the *GNAQ* gene was highly expressed in the hypothalamus of Kazakh sheep and that the activity of the hypothalamus–pituitary–gonadal axis (HPGA) was influenced by photoperiod, which was then treated with different concentrations of folic acid in the hypothalamic neuronal cells of Kazakh sheep. The results showed that appropriate folic acid concentrations promoted methylation of the *GANQ* promoter and affected GnRH secretion [13]. In addition, the upregulation of *GNAQ* gene expression caused a significant acceleration in the proliferation rate of SH-SY5Y cells and significantly upregulated NF-κB expression, thereby inhibiting GnRH neuronal cell proliferation through the NF-κB signaling pathway [14,15,16].

The main epigenetic mechanisms reported are DNA methylation, post-translational modifications of histones, and non-coding RNAs that can control gene expression, and previous studies have demonstrated that epigenetic changes play an important role in biological rhythms [17]. Stevenson found that the pharmacological blockade of photoperiod-driven demethylation attenuated the reproductive response to the winter photoperiod in their study [18]. That is photoperiodic- and MEL (melatonin)-driven changes in reproduction and hypothalamic *Dio3* (type III iodothyronine deiodinase) expression are controlled by Dio3 proximal promoter methylation, and neuroendocrine refractoriness during rumbling can be mediated with the reversal of DNA methylation patterns [18,19]. Furthermore, the histone deacetylase 3 (HDAC3)/nuclear corepressor 1 (Ncor1) complex requires rhythmic and reversible deacetylation of histone H4 for proper circadian clock functional expression. Mutations that block the HDAC3-related deacetylation activation domain of Ncor1 significantly shortened the circadian cycle length and dysregulated circadian clock gene expression of *BMAL1* (brain and muscle Arnt-like protein 1) and Rev-ERbα [20]. In honeybees, the reversible transition from a “mammalian” to a “foraging” behavioral phenotype is associated with genomic changes in methylation patterns [21]. In addition, it has been shown that specific non-coding RNAs can regulate candidate genes associated with seasonal estrus in sheep [22,23].

So far, most studies have focused on the significance of *GNAQ* in human diseases and model animal signaling, but reports in the study of reproductive traits in mammals such as domestic animals are still limited, and specific molecular mechanisms need to be investigated in depth. It has been found that elevated *GNAQ* gene expression following miR-200b interference leads to significant upregulation of GnRH [24]. Although large-scale “methylome” data suggest that high levels of cytosine methylation occur at the CpG islands of promoters of important reproductive genes (e.g., the *Dio3* gene), little research has been carried out on *GNAQ*. Therefore, in this study, we analyzed the DNA methylation level of CpG islands near the core promoter region of the *GNAQ* gene, and whether the expression of GNAQ after histone acetylation would affect seasonal reproduction to explore whether photoperiodic variations would induce DNA methylation, which in turn would lead to significant differences in the expression of genes or proteins, based on elucidating the differences in gene expression under different photoperiodic conditions. The results may provide new epigenetic insights into the photoperiodic response (seasonal estrus) of sheep.

## 2. Results

### 2.1. The DNA Methylation in the GNAQ Gene using Whole Genome Methylation Sequencing (WGBS)

After WGBS analysis, the results of differentially methylated regions (DMRs) and loci between different groups in the *GNAQ* gene are shown in Figure 1. Figure 1A shows the DMRs between SP42, LP42, and SP-LP42. Figure 1B is a zoom of “I” in Figure 1A. Figure 1B indicates the differentially methylated regions between SP42 and LP42. Figure 1C,D show the enlargement of “II and III” in Figure 1A, indicating the different methylation sites between SP42 and SP-LP42.

### 2.2. The Expression of Major Methylated Transferase Genes in the Different Photoperiods

To investigate whether photoperiod changes lead to DNA methylation of the *GNAQ* gene, we first analyzed the expression of major methylation transferase genes (*DNMT1*, *DNMT3A,* and *DNMT3B*) under different photoperiods using qRT-PCR. The results are shown in Figure 2. In the sheep hypothalamus, *DNMT1* showed significantly higher expression in SP42 than in LP42 (*p* < 0.05); *DNMT3A* showed significantly higher expression in SP42 than SP-LP42 (*p* < 0.01), and its expression in LP42 was significantly higher than that in SP-LP42 (*p* < 0.05); and *DNMT3B* showed significantly higher expression in SP42 than in LP42 and SP-LP42 (*p* < 0.05). This implies that gene expression in the sheep hypothalamus may be regulated by DNA methylation under different photoperiods.

### 2.3. GNAQ Gene Core Promoter Identification

To study the activity of the sheep *GNAQ* gene promoter, we constructed a series of deletion luciferase reporter constructs and transfected them with HEK293T cells. The luciferase activity of *GNAQ* promoters of different sizes was measured using a dual luciferase reporter system. As shown, the luciferase activities of the luciferase reporter constructs in HEK293T cells were higher than those of the pSI-basic negative control (Figure 3), but pSI-P4 showed a significant increase in luciferase activity compared with pSI-basic and pSI-P3 (*p* < 0.05); pSI-P4 and pSI-P5 promoter regions had higher luciferase activity than pSI-basic. pSI-basic is not significantly different to pSI-P1, pSI-P2, and pSI-P3. The results suggest that the *GNAQ* promoter core region is located from −1100 to −1500 bp.

### 2.4. The DNA Methylation Verification of the DMR in the GNAQ Gene

We verified the DNA methylation level of each CpG site of *GNAQ* except the core promoter region with pyrosequencing, and the results are shown in Figure 4. The sequence and the location of CpG sites for pyrosequencing in the three DMRs from the WGBS result are shown (Figure 4A,C,E). After amplification of this region, there are 13 different DNA methylation CpG sites among SP42, LP42, and SP-LP42 (Figure 4), which we named CpG sites 1–5 (CpG 4B1–4B5/4D1–4D3/4F1–4F5) from left to right, respectively. The DNA methylation levels of five CpG sites in the DMR of SP42 and LP42 were not significantly different (*p* > 0.05) (Figure 4B). The DNA methylation level in the DMR of the CpG1 site in SP42 was significantly higher than that in SP-LP42 (*p* < 0.01) (Figure 4D). The expression of *GNAQ* in SP42 was lower than that in SP-LP42, suggesting that methylation of CpG 4D1 in the promoter region of *GNAQ* may affect the expression of *GNAQ.* The DNA methylation levels of the remaining seven CpG sites in the DMR of SP42 and SP-LP42 were not significantly different (*p* > 0.05) (Figure 4D,F).

### 2.5. The DNA Methylation Analysis of the CpG Island in the Core Promoter Region

We used pyrosequencing to analyze the DNA methylation levels of each CpG site in the core promoter region. The results of pyrosequencing two sequences in the core promoter region are shown in Figure 5A,C. There are eight different DNA methylation CpG sites in the core promoter region among SP42, LP42, and SP-LP42, which we named CpG sites 1–8 (CpG 5B1–5B4/5D1–5D4) from left to right, respectively. The DNA methylation levels of all CpG sites in the core promoter among three different photoperiod treatments for each region were significantly different (Figure 5). The DNA methylation level of the CpG sites in the core promoter region in SP42 was significantly higher than that in LP42 (*p* < 0.01), and significantly higher than that in SP-LP42 (*p* < 0.01), except CpG 5B2 and CpG 5B4 in the first fragment (*p* < 0.05) (Figure 5B,D). This suggests that DNA methylation may be responsible for the differential expression of the *GNAQ* gene under different photoperiods.

### 2.6. Hypothalamic Histone H3 Acetylation under Different Photoperiods

The acetylation status of *GNAQ* histone H3 under different photoperiods was assessed using chromatin immunoprecipitation (ChIP)-qPCR. As shown in Figure 6, the expression of acetylated *GNAQ* histone H3 was significantly higher than that of the control group under three different photoperiods (*p* < 0.01) (Figure 6A); the acetylation level of sheep hypothalamic *GNAQ* genomic protein H3 was significantly lower under SP42 than under SP-LP42 (*p* < 0.05) (Figure 6B). This suggests that acetylated histone H3 binds to the core promoter region of the *GNAQ* gene, implying that *GNAQ* is epigenetically regulated by photoperiod through histone acetylation.

### 2.7. The Expression Differences of GNAQ in Different Photoperiods

The gene and protein expression of GNAQ was different in the hypothalamus under different photoperiods. The expression of GNAQ mRNA was not significantly higher in SP42 than in LP42; GNAQ mRNA expression in SP-LP42 was significantly different from that in LP42 and SP. (Figure 7A). Western bolt and grayscale analysis showed the expression of GNAQ in protein was consistent with the trend of mRNA, and there was a significant difference in the group of SP-LP42 vs. LP42 (*p* < 0.05), while the expression of GNAQ did not show a significant difference in the group of SP42 vs. LP42 and SP42 vs. SP-LP42 (Figure 7B,C).

## 3. Discussion

The seasonal estrus of ewes restricts the balanced production of sheep throughout the year. In recent years, researchers have devoted themselves to the study of the molecular regulation mechanism of seasonal estrus in sheep, aiming to analyze the mechanism at the molecular level. Using high-density genotyping techniques, Yurchenko et al. conducted the first comprehensive scan of genomic selection characteristics of European- and Asian-origin sheep breeds from the Russian Federation and determined that *GNAQ* may be a major gene influence on reproductive traits in Russian sheep [25]. Earlier studies suggest that *GNAQ* may be a major gene controlling seasonal estrus in sheep [26]. Downregulation of *GNAQ* expression regulates kisspeptin expression and indirectly regulates the *GnRH* gene through the kisspeptin/GPR54 signaling pathway, ultimately upregulating GnRH hormone levels and regulating estrus in sheep through HPG [27]. However, current research on *GNAQ* focuses on cancers and autoimmune diseases, while seasonal estrus in sheep has not been well studied [28,29,30]. The hypothalamus is located upstream of the HPGA and its secreted GnRH is the main regulator of fertility in animals. *GNAQ* has a high expression level in hypothalamic tissues, suggesting that *GNAQ* may have a regulatory effect on the expression of hypothalamic GnRH [27]. In this study, we analyzed the expression of *GNAQ* in Sunite sheep under different photoperiods, and the expression of *GNAQ* mRNA was significantly higher in SP-LP42 than in LP42. *GNAQ* mRNA expression in SP42 was not significantly different from that in LP42 and SP-LP42. This is consistent with the previous results and suggests that *GNAQ* may be associated with the seasonal estrus of sheep.

Studies have shown that epigenetic modifications play an important role in animal reproduction [31,32,33], where epigenetic repression mechanisms targeting genes required for pubertal activation of GnRH neurons have been identified as a central component of the molecular mechanism of central inhibition of puberty [34]. Epigenetic modifications mainly consist of (a) chemical modification of DNA through DNA methylation and hydroxymethylation; (b) structural modification of chromatin caused by post-translational modification (PTM) of histones, which are wrapped by two superhelical spins of DNA and constitute nucleosomes, the core units of chromatin; and (c) non-coding RNAs and RNA modification [35,36,37]. Yang et al. screened and validated the differential expression of miR-200b-GNAQ pairs. A dual-luciferase reporter gene assay showed that miR-200b could bind to the 3′-untranslated region of *GNAQ* to mediate the hypothalamic–pituitary–ovarian axis, thereby affecting seasonal estrus in sheep [24]. Later studies also demonstrated that the *GNAQ* gene received regulation via non-coding RNAs [38,39].

Whether the *GNAQ* gene is affected by DNA methylation and histone modifications is less studied. Studies have used isolated murine sperm cells and revealed that DNA methylation and histone 3 Lys-9 trimethylation (H3K9me3) have different sequence preferences and dynamics in promoters and repetitive elements during spermatogenesis. H3K9me3 modifications in gene promoter histones were highly enriched in round spermatozoa [40]. Also, during immunostaining of young (3 months old) and old (12 months old) mouse testes to qualitatively and semi-quantitatively assess the pattern of histone modifications in male spermatogenic cells during spermatogenesis, it was found that some histone modifications differ in intensity and that histone modifications on sex chromosomes and other chromosomes are differently regulated by aging [41]. Lomniczi et al. showed that female primiparity is epistatically regulated, and the binding status of the kisspeptin 1 (kiss1) gene promoter region to embryonic ectodermal development proteins is influenced by methylation and acetylation [35,42]. The kiss1 gene promoter region is influenced by methylation and acetylation, which ultimately alter female primiparity in rats [9,11]. This suggests that DNA methylation and histone modifications critically regulate the expression of many genes and repetitive regions during spermatogenesis. In our study, the mean methylation of the promoter region of the *GNAQ* gene differed significantly under different photoperiodic states in Sunite sheep and was significantly and negatively correlated with the mean expression, indicating that the differential expression of the *GNAQ* gene in the hypothalamus of sheep is likely to be regulated by DNA methylation in the promoter region of this gene. This is consistent with the previous results. In contrast, downregulation of *GNAQ* gene expression promotes GnRH secretion, which in turn affects seasonal estrus in animals, which is consistent with previous results. The Japanese quail is a truly photoperiodic avian species. It was found that prolonged exposure of Japanese quail to long light reduced photosensitivity, while prolonged exposure to short-light conditions led to the development of avian photosensitivity, which then resulted in increased activity of the HPGA [43].

In this study, GNAQ was expressed at a lower level in LP42, but at a lower level of DNA methylation. This suggests that there may be a reduction in photosensitivity due to prolonged exposure to long light in sheep, or there may be highly expressed repressors that bind to the DNA promoter region under the light that may weaken DNA methylation and thus inhibit GNAQ expression. After the chromatin immunoprecipitation assay of hypothalamic tissues in this study, acetylated histone H3 was found to bind to the core promoter region of the *GNAQ* gene under different photoperiods, which suggests that *GNAQ* is subjected to epigenetic regulation of photoperiods through histone acetylation. The level of histone H3 acetylation of the hypothalamic *GNAQ* gene of sheep was significantly lower under SP42 than under SP-LP42, which indicates that histone H3 acetylation plays a major role during sheep overbreeding, consistent with what was described by previous authors. The above results indicate that photoperiod can induce DNA methylation in the core promoter region and histone acetylation in the promoter region of the *GNAQ* gene. Our results allow us to speculate that the two may be key factors in regulating differential expression of *GNAQ* under different photoperiods, and thus regulating the seasonal estrus in sheep through the HPGA. In addition, mammalian gene DNA methylation is based on the DNA methylation transferases *DNMT3A* and *DNMT3B* [44,45]. Lynch found that the methylation transferase DNMT3A gene regulates DNA methylation in the testis, ovary, and uterus of adult Siberian hamsters [46]. The expression of *DNMT3A* was influenced by photoperiod, MEL, and ovarian hormones, and the expression of *DNMT3A* affected reproductive activity according to the season [46]. We also detected the expression levels of the major DNA methylation transferases (*DNMT1, DNMT3A, DNMT3B*). The results showed that DNMT3A and DNMT3B were highly significantly differentially expressed in SP42 and SP-LP42 (*p* < 0.01), which was consistent with the sequencing results. This suggests that the degree of methylation in the promoter region of the *GNAQ* gene is positively regulated by methylation transferase.

## 4. Materials and Methods

### 4.1. Animals and Sample Collection

Nine clinically normal and non-pregnant Sunite ewes (35–40 kg, 3 years old) from Urat Middle Banner, Bayan Nur City, Inner-Mongolia Autonomous Region, China, were selected and kept at the farm of the Tianjin Institute of Animal Science, located in Tianjin (39° N latitude), China. The ewes were reared on ad libitum feeding and water intake basis. Sheep were treated with bilateral oophorectomy and estradiol, with procedures described in our previous studies [8]. Ewes were equally assigned to three rooms with controlled light conditions: SP: short photoperiod with lights on for 8 h (10:30–18:30) and off for 16 h (18:30–10:30); LP: long photoperiod with lights on for 16 h (6:30–22:30) and off for 8 h (22:30–6:30); SP-LP: short photoperiod to long photoperiod. All ewes were slaughtered at SP42 days, LP42 days, and SP-LP42 days, respectively, and the hypothalamic tissues removed were washed with PBS (pH 7.4), snap-frozen in liquid nitrogen, then stored at −80 °C for subsequent studies.

All the experimental procedures mentioned in the present study were approved by the Science Research Department (in charge of animal welfare issues) of the Institute of Animal Sciences, (IAS-CAAS) (Beijing, China). Ethical approval was given by the Animal Ethics Committee of the IAS (IAS2021-24).

### 4.2. DNA Isolation and Bisulfite Treatments

Genomic DNA was extracted from each sample using the phenol-chloroform method and dissolved in ddH_2_O after extraction, using the Epitect Bisulfite kit (Qiagen, Valencia, CA, USA) according to the manufacturer’s manual. Briefly, a 140 μL reaction system containing 1 μg DNA (20 μL), 85 uL Bisulfite Mix, and 35 uL DNA Protect Buffer was used for conversion. After chemical conversion, unmethylated cytosine was converted to uracil, methylated cytosine was protected, and the converted DNA was extracted for subsequent DNA methylation analysis.

### 4.3. Whole Genome Methylation Sequencing (WGBS)

After bisulfite treatment, the sample library was equilibrated, mixed with other libraries of different barcodes, sequenced on different HiSeq XTen tracks (Illumina, San Diego, CA, USA), and Novogene paired (Novogene, Beijing, China) end of 150 bp reads were created. Purification data from each sample were collected using bowtie2 with Bismark-0.14.5 software and compared to the reference genome (*Oar_v4.0*) to determine post-sequencing parameters. Methyl cytosine information was extracted after duplicate extraction of duplicate reads using the Bismarck methylation extractor, ignoring the first 5 bp of paired-end reads to reduce the effect of the strong bias of unmethylated-end reads due to end recovery. The results of the methylation extractor were converted to bigwig format so that they could be viewed with the Integrative Genomics Viewer (IGV). To confirm methylation, a binomial distribution test was performed for each C site under the screening conditions with at least 5-fold coverage and false discovery rate (FDR) ≤ 0.01. To identify DMRs between groups, DMR analysis was performed using MOABS for all C sites with >5-fold read coverage. DMRs were then identified in regions with Fisher’s exact test, *p*-value less than 0.05 if the methylation difference was greater than 0.2 (CG background was greater than 0.3), and at least three methylation sites were present.

### 4.4. Isolation of 5′-Flanking Region and Luciferase Reporter Vector Construction

We used Promoter-2.0 (https://services.healthtech.dtu.dk/service.php?Promoter-2.0, accessed on 23 October 2021) to predict the putative core promoter. The 2021 bp upstream of the transcription start site (TSS) of the *GNAQ* gene sequence was obtained using the PCR-driven overlap extension method. Different lengths of promoter region vector deletion were constructed by ensuring that the region near the TSS end was not moved and subtracting different length bp sequences from the other end, respectively, forming P1 (−400 to +21), P2 (−700 to +21), P3 (−1100 to +21), P4 (−1500 to +21), and P5 (−2000 to +21) fragments. pSICheck-2 basic vector was double enzymatically excised with Xho I and Not I, and the excised vector was connected to the above-amplified fragments. The connection system comprised the following: linear vector 100 ng, 2 × SoSoo Mix 5 μL, target fragment 100 ng, and ddH_2_O to supplement to 10 μL. The vectors were connected at 50 °C for 30 min and sent to the company for sequencing after successful ligation.

### 4.5. Cell culture, Transfections, and Luciferase Assay

The 293t cells were maintained in a medium (DMEM, Gibco, Beijing, China) supplemented with 10% fetal bovine serum (FBS, Gibco) and incubated at 37 °C under 5% CO_2_. Before transfection, 293 cells were plated at 0.5 × 105 cells/well in 24-well plates for culture overnight. Subsequently, the transient transfections were performed with the promoter luciferase reporter constructs using Lipofectamine™ 2000 (Invitrogen, Carlsbad, CA, USA). A promoter luciferase reporter plasmid or a positive control plasmid pSICheck-2-control vector (Promega, Madison, WI, USA) was used for each well, while the ratio of total plasmid DNA to Lipofectamine™ 2000 transfection reagent was 1:2.5. Cells were transfected in triplicate for each recombinant plasmid. After 24 h transfection, the cells were lysed, and Firefly and Renilla luciferase activities were analyzed with Dual-Luciferase^®^ Reporter Assay System (Promega), while the luminescence was determined on Tecan Infinite^®^ 200 Pro (Tecan Group LTD, Shanghai, China). Firefly luciferase activities were normalized with the Renilla luciferase activity (pRL-TK) in each well, and the observed values were compared with the value of the negative control luciferase vector pSI-Basic.

### 4.6. Pyrosequencing Analysis

A pyrosequencing protocol was employed to verify the results of WGBS and to analyze the DNA methylation status of the core promoter of the GNAQ gene. Pyrosequencing amplification and sequencing primers were designed by Assay Design Software in Pyro Mark Assay Design 2.0 (Qiagen, Valencia, CA, USA). The primer information and product sizes of the fragments are shown in Table 1. The PCR reaction was performed in a volume of 25 μL according to the Pyro Mark^®^ PCR kit, containing 2.5 μL 10× PCR Cora Load Concentrate, 5 μL of 5× Q-solution, 0.5 μL of each primer (10 μM), 12.5 μL 2× Pyro Mark PCR Master Mix, and 50 ng bisulfite-treated genomic DNA. The PCR amplification conditions were as follows: denaturation at 95 °C for 15 min, then 45 cycles of 94 °C for 30 s, optimal annealing temperature for each specific primer for 30 s, 72 °C for 30 s, and a final hold of 20 °C. Subsequently, the pyrosequencing primer was used in the sequencing of the PCR products. The sequencing reaction was conducted in the PyroMark Q96 ID system (Qiagen, Valencia, CA, USA) with the PyroMark Gold Q96 Reagents kit (Qiagen, Valencia, CA, USA) according to the manufacturer’s protocols.

### 4.7. Chromatin Immunoprecipitation-qPCR (ChIP-qPCR)

Chromatin immunoprecipitation (ChIP) was performed as described previously [47,48]. Briefly, crosslinking was performed with 1% formalin, the cells were lysed in SDS buffer, and sonication was used to fragment the DNA. ChIP for GNAQ was performed using a Flag antibody (Sigma, SAB4301135, St. Louis, MO, USA). Eluted DNA fragments were analyzed using qPCR. The primers are listed in Table 2.

### 4.8. Quantitative Real-Time PCR (qPCR)

Total RNA was isolated from each sample using TRIzol reagent (Invitrogen, Carlsbad, CA, USA), and 1% agarose gels were used to detect the degradation and contamination before the subsequent study. cDNA was synthesized from the RNA samples remaining after sequencing with the PrimeScript™ RT reagent Kit (TaKaRa, Dalian, China). The primers were designed for qPCR in our laboratory and synthesized by Sangon (Shanghai, China). The PCR mixture contained 2.0 μL cDNA, 10.0 μL SYBR Real-time PCR System, and 0.8 μL forward and reverse primers (10 μM). Cycle conditions were 95 °C for 10 mins, followed by 40 cycles of 95 °C for 15 s, 60 °C for 1 min, and 95 °C for 15 s. GAPDH was used as an internal control for mRNA quantification. The mRNA expression levels were determined using the 2^−ΔΔCt^ method. Three biological replicates for each transcript were used, and the primers are listed in Table 2.

### 4.9. Western Blot Analysis

Hypothalamus tissue was grounded to a fine powder in liquid nitrogen. Tissues and cells were lysed in RIPA with 1% benzoyl fluoride (PMSF) (Beyotime, Shanghai, China) for 1–2 h, then centrifuged at 12,000 rpm for 30 min at 4 °C, and the resulting supernatant was assayed for concentration with bicinchoninic acid (BCA) assay. A 10% gel was prepared with sodium dodecyl sulfate for separation, and a concentrated gel was prepared. The proteins were transferred to nitrocellulose membranes at 80 V for 1 h and 120 V for 0.5 h. The stained material was washed three times and incubated with a secondary antibody. After washing, the membranes were stained using the ECL Western Transfer Kit (Beyotime, Shanghai, China). Finally, color development was performed with a western blot imaging instrument. The following antibodies were used: rabbit Anti-IgG (1:5000, Proteintech, Chicago, IL, USA); goat anti-GNAQ (1:1000, Abcam, MA, USA), and rabbit anti-GAPDH (1:2000, Proteintech, Chicago, IL, USA). The bands of interest in the western blots were normalized to GAPDH.

### 4.10. Statistical Analysis

Statistical evaluation of the data used SPSS (v.22). A one-way ANOVA test was used for statistical analysis. The Pearson correlation coefficient was used to calculate the correlation between DNA methylation and gene expression, and the significance was also calculated. The data were presented as mean ± standard error (SE) values of independent determinations. *p* < 0.05 and *p* < 0.01 were considered statistically significant or highly significant.

## 5. Conclusions

In summary, we found that the core promoter of *GNAQ* is located 1000–1500 bp before the transcription start site. The results suggest that photoperiod can induce DNA methylation in the core promoter region and histone acetylation in the promoter region of the *GNAQ* gene and hypothesize that the two may be key factors in regulating the differential expression of *GNAQ* under different photoperiods, thus regulating the HPGA throughout seasonal estrus in sheep. Our studies revealed the effects of photoperiod on gene expression and ovine reproductive activity from the perspective of epigenetics, and these findings provide a new idea for studying seasonal estrus and reproduction in sheep.

## Figures and Tables

**Figure 1 ijms-24-16442-f001:**
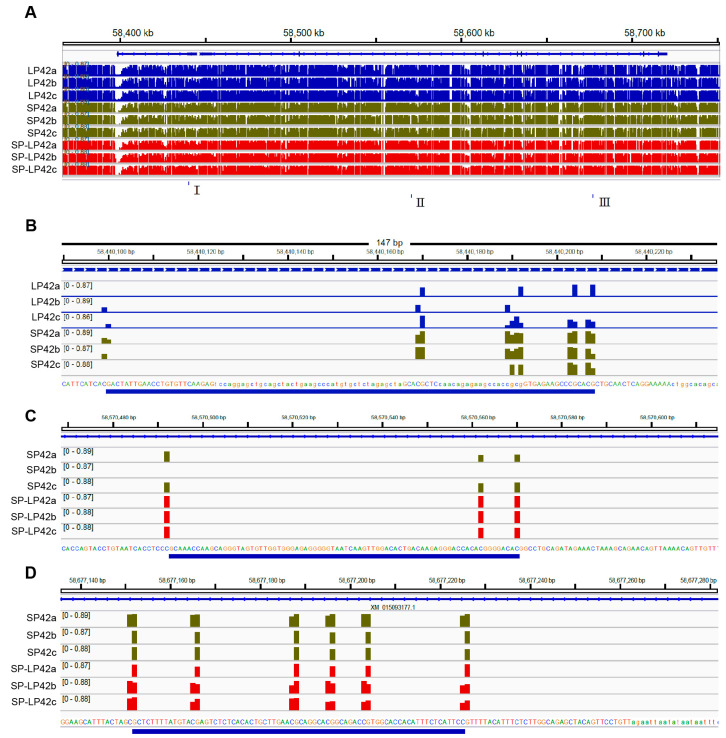
The differentially methylated regions (DMRs) and sites between the different groups in the *GNAQ* gene. (**A**) The DMRs between SP42, LP42, and SP-LP42. (**B**) The enlargement of “I” in (**A**), represents the different methylated sites between SP42 and LP42. (**C**,**D**). The enlargement of “II and III” in (**A**), represents the different methylated sites between SP42 and SP. The expression of major methylated transferase genes in the different photoperiods LP42.

**Figure 2 ijms-24-16442-f002:**
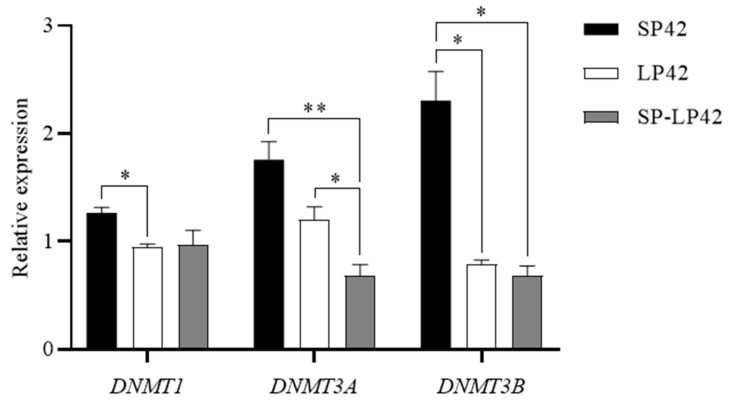
The expression of major methylation transferase genes (*DNMT1*, *DNMT3A*, and *DNMT3B*) under different photoperiods. * *p* < 0.05, ** *p* < 0.01.

**Figure 3 ijms-24-16442-f003:**
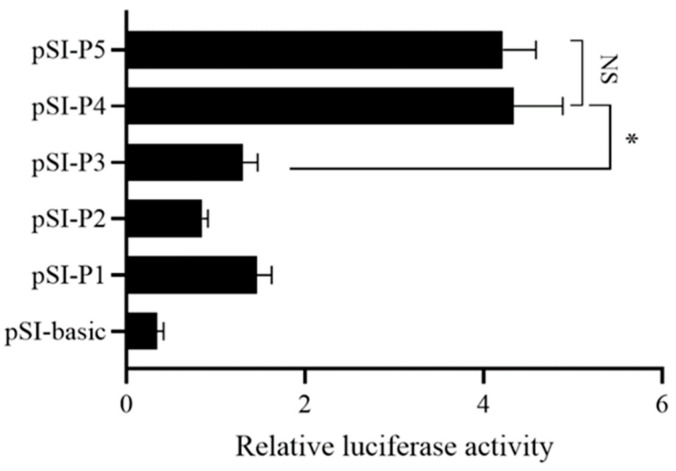
Luciferase assay. The deletion constructs were transfected into HEK293T cells. The results are expressed as the mean ± SEM (*n* = 3) in arbitrary units based on firefly luciferase activity normalized against Renilla luciferase activity. The results are an average of three independent experiments performed in triplicate. * *p* < 0.05; NS: no significance.

**Figure 4 ijms-24-16442-f004:**
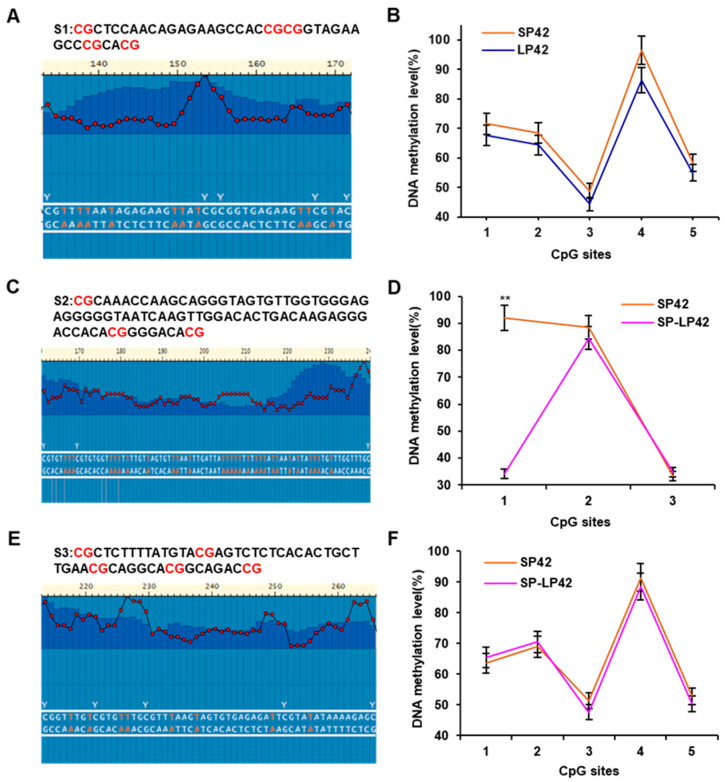
The pyrosequencing verification of DNA methylation degree of each CpG site. (**A**,**C**,**E**) The sequence and the location of CpG sites for pyrosequencing in the three DMRs from the WGBS result. (**B**) Comparison of the DNA methylation levels of 5 CpG sites (CpG 4B1–4B5) in DMR between SP42 and LP42. (**D**) Comparison of the DNA methylation levels of 3 CpG sites (CpG 4D1–4D3) in DMR between SP42 and SP-LP42. (**F**) Comparison of the DNA methylation levels of 5 CpG sites (CpG 4F1–4F5) in DMR between SP42 and SP-LP42. S1-S3 represents the sequence of DMR screened using WGBS, and the red “CG” indicates a possible methylation site. The significance is expressed as a comparison with SP42, ** *p* < 0.01.

**Figure 5 ijms-24-16442-f005:**
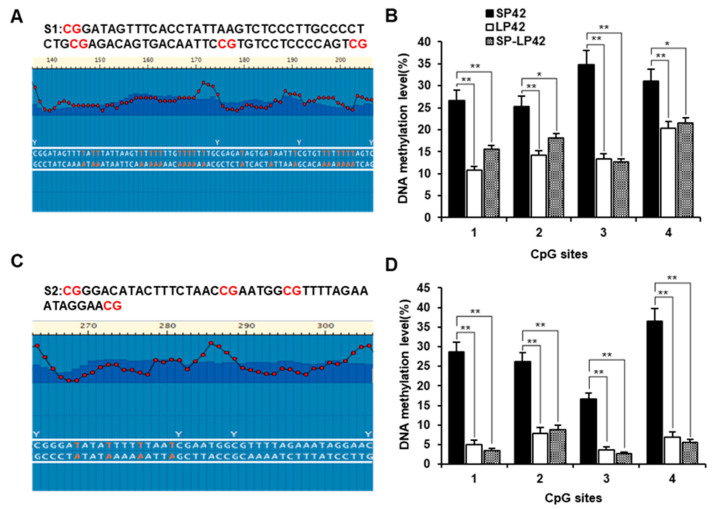
The pyrosequencing analysis of the DNA methylation level of each CpG site in the core promoter region. (**A**,**C**) Two sequences for pyrosequencing in the core promoter region. (**B**,**D**) Comparison of the DNA methylation levels of 4 CpG sites (CpG 5B1–5B4/5D1–5D4) in the core promoter among the three different photoperiod treatments in each region. * *p* < 0.05, ** *p* < 0.01.

**Figure 6 ijms-24-16442-f006:**
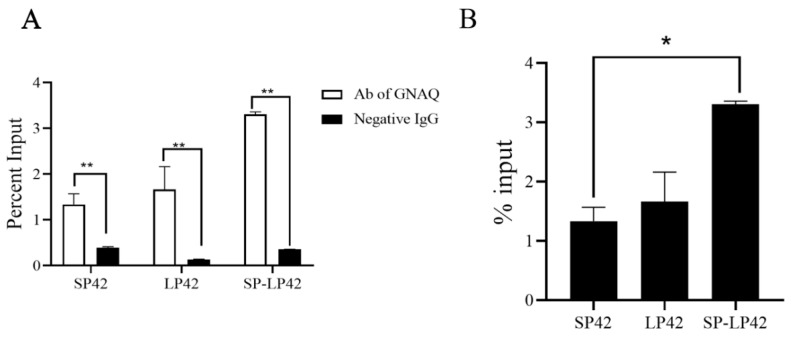
The chromatin immunoprecipitation (ChIP)-qPCR of the *GNAQ* in the hypothalamus among the different photoperiod treatments. (**A**) ChIP assay shows that acetyl-histone H3 binds to the *GNAQ* promoter. (**B**) Histone acetylation of *GNAQ* under different photoperiods. (* *p* < 0.05, ** *p* < 0.01).

**Figure 7 ijms-24-16442-f007:**
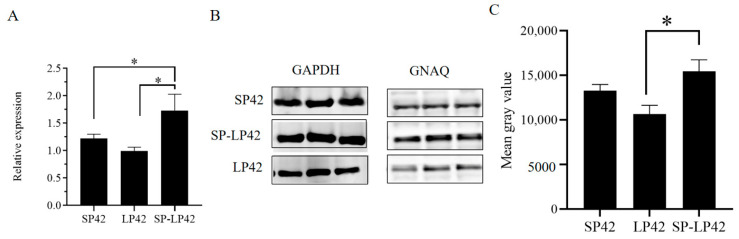
The expression of GNAQ in the hypothalamus among the different photoperiod treatments. (**A**): q-PCR analysis of the efficiency of GNAQ in the different photoperiods. (**B**): Weston blot analysis of the efficiency of glyceraldehyde-3-phosphate dehydrogenase (GAPDH) and GNAQ in the different photoperiods. (**C**): Relative expression of GNAQ using protein grayscale analysis. * *p* < 0.05.

**Table 1 ijms-24-16442-t001:** The primer information for pyrosequencing.

Location	Primer Name	Sequences (5′-3′)	Product Size (bp)
DMR	GNAQ-F1	TGATTTTTGGTTTGGGAAGATTTTAT	203
	GNAQ-R1	AATCTTTATTACTATACCAATTTTTCCT	
	GNAQ-S1	AGTTTATGTGTTTTAGAGTTAG	
	GNAQ-F2	TTAGTGTTATTTGGGAGGATTATATTAGG	198
	GNAQ-R2	CACTACATACTACCACCAATACCTA	
	GNAQ-S2	CCAATACCTATAATCACCT	
	GNAQ-F3	GGAATTGTAGTTTTGTTAAGAGAAATGTAA	136
	GNAQ-R3	ATCCACTTCCAAAAAAAACATTTACTA	
	GNAQ-S3	GGAATGAGAAATGTGGT	
Core promoter	GNAQ-F1	GGTATAAAAAGTTGGAAGTTAGTAGG	269
	GNAQ-R1	AAATATATCCCCTCACCTCTAATCCAAATT	
	GNAQ-S1	ATAATATACACTCAACTATACAAT	
	GNAQ-F2	GGTATAAAAAGTTGGAAGTTAGTAGG	329
	GNAQ-R2	ACTCTTCCCCTAATTCAATATTCTTTCC	
	GNAQ-S2	ATTATTTAATTTGGATTAGAGGT	

**Table 2 ijms-24-16442-t002:** The primer information for qPCR.

Usage	Primer Name	Sequences (5′-3′)	Product Size (bp)
qPCR	GNAQ-F	GGACAGGAGAGAGTGGCAAG	127
	GNAQ-R	TAGGGGATCTTGAGCGTGTC	
	ACTB-F	GCTGTATTCCCCTCCATCGT	97
	ACTB-R	GGATACCTCTCTTGCTCTGG	
	DNMT1-F	AGCCCCAGTCTTGGTTCCA	85
	DNMT1-R	GCGCTCATGTCCTTGCAAAT	
	MGB-Probe	CCATCCTCAGGGATC	
	DNMT3A-F	CGTCTCGGCTCCAGATGTTC	59
	DNMT3A-R	CTTCGGAGGGTCGAATTCCT	
	MGB-Probe	CGCCAACAACCATG	
	DNMT3B-F	ACCGACGGCGGCCTAT	59
	DNMT3B-R	CAAGTACCCTGTTGCAATTCCA	
	MGB-Probe	CGAGTCTTGTCGCTGTT	
	ACTB-F	GCAGCCAAAAGCATCACCAA	114
	ACTB-R	TCACCGGAGTCCATCACGAT	
	MGB-Probe	GCAGCCAAAAGCATCACCAA	
ChIP + qPCR	F	GTGTCCTCCCCAGTCGCA	
	R	CGGCTCTTCGCCTAGTTCAG	

## Data Availability

No new data were created or analyzed in this study. Data sharing is not applicable to this article.

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
