# Peer review of "Photoperiod Induces the Epigenetic Change of the GNAQ Gene in OVX+E2 Ewes"

_ijms, 2023, doi:10.3390/ijms242216442_

Round 1
Reviewer 1 Report
Comments and Suggestions for Authors
Seasonal regulation of animal reproduction in temperate climate zones is a very complex process that has been of interest to researchers since the discovery of the neurohormone GnRH and the importance of melatonin signaling. After the identification of further neuronal networks, including NPY, KNDy neurons, etc., research at the molecular level became increasingly important. Understanding the sheep genome has made it possible to track the expression of many genes during the reproductive cycle throughout the year, and now more and more attention is also paid to epigenetics. The authors of this study fit into the trend of the latter research and set the goal of examining, among other, the DNA methylation levels of the GNAQ gene, and the acetylation status of GNAQ histone under different photoperiods. In short, GNAQ encodes for Gαq, a guanine nucleotide protein which is part of a trimeric G protein complex that couples with certain protein coupled receptors. It is highly expressed in the hypothalamus and may affect the hypothalamic-pituitary axes.
The work submitted for evaluation is extremely interesting and contains elements of modern science, linking the latest discoveries at the molecular level with known life functions that depend on the season. However, it contains a few imperfections that should be corrected.
1. Several abbreviations are explained in the materials and methods section, but appear earlier, e.g., HPG, line 36; GnRH, line 51; GPR54, line 56; WGBS, line 111/112; DMRs, line 112; SP43, LP42 and SP-LP42, Results section; Kiss1, line 265; GAPDH, line 413 and Figure 7; What is the difference between HPG axis and HPGA, line 230 and 233?
2. Introduction, line 102-107: A long, complex sentence is not understandable. State specific hypotheses or objectives, preferably separately.
3. Can the authors include in the Discussion information about the endogenous reproductive rhythm of sheep, which is revealed under constant photoperiodic conditions, and whether changes in GNAQ gene methylation can occur under such conditions?
4. It is not clear what degree of methylation is during the induction of the reproductive period in sheep and what is associated with the anestrous period?
5. Add another number to the Materials and Methods section.
6. Statistical analysis: explain the shortcut SPSS.
7. Conclusions (point 5) can be moved to the end of the Discussion section (before Materials and Methods).
Author Response
- Several abbreviations are explained in the materials and methods section, but appear earlier, e.g., HPG, line 36; GnRH, line 51; GPR54, line 56; WGBS, line 111/112; DMRs, line 112; SP43, LP42 and SP-LP42, Results section; Kiss1, line 265; GAPDH, line 413 and Figure 7; What is the difference between HPG axis and HPGA, line 230 and 233?
Response: Thank you for the reminder, we have revised the article. there is no difference between HPG axes and HPGA, so we have standardized the article to use "HPGA".
- Introduction, line 102-107: A long, complex sentence is not understandable. State specific hypotheses or objectives, preferably separately.
Response: Thank you for your suggestion, we have made changes to it.
- Can the authors include in the Discussion information about the endogenous reproductive rhythm of sheep, which is revealed under constant photoperiodic conditions, and whether changes in GNAQ gene methylation can occur under such conditions?
Response: Thank you for your suggestion, we apologize that we don't have a record of this part of the information and cannot provide it.
- It is not clear what degree of methylation is during the induction of the reproductive period in sheep and what is associated with the anestrous period?
Response: Thank you for your suggestion, we will follow up on this part of the work.
- Add another number to the Materials and Methods section.
Response: Thank you for your suggestion, we have modified it.
- Statistical analysis: explain the shortcut SPSS.
Response: SPSS (Statistical Product and Service Solutions). It is a statistical analysis software.
- Conclusions (point 5) can be moved to the end of the Discussion section (before Materials and Methods).
Response: Thank you for the suggestion, we put the "Conclusion" section here in the journal format.

Reviewer 2 Report
Comments and Suggestions for Authors
The authors present a study on the expression and epigenetic changes of GNAQ in the hypothalamus in ovariectomized and estradiol-treated (OVX+E2) sheep under three different photoperiod treatments for 42 days (short, long and short-long, SP42, LP42, SP42-LP42).
Expression of GNAQ was significantly higher in SP42-LP42 than in SP42 and LP42.
Whole genome methylation sequencing (WGBS) revealed multiple differentially methylated regions (DMRs) and loci.
DNA methylation level of DMRs at the CpG1 locus in SP42 was significantly higher than that of SP-LP42.
DNA methylation level of all 8 CpG sites in SP42 was significantly higher than that in LP42 and SP-LP42 (p < 0.01), except site 2 and site 4 in the first sequencing fragment of the core promoter region.
Expression of acetylated GNAQ histone H3 was significantly higher than that of the control group under three different photoperiods.
Acetylation level of sheep hypothalamic GNAQ genomic protein H3 was significantly lower under SP42 than under SP-LP42.
Results suggest an epigenetic regulation of GNAQ and that photoperiod may induce DNA methylation in the core promoter region and histone acetylation in the promoter region of GNAQ.
Comments
Propose that authors provide some more data on genes influencing season in sheep and the HPG axis. This way the reader can get a better impression on the potential role of GNAQ. A supplementary table would be useful.
Lines 136-145: please add fragment length for pSI-P1 to pSI-P5.
Line 152: We verified the DNA methylation level of each CpG site of GNAQ by pyrosequencing, and the results are shown in Figure 4.
and in Line 174: We used pyrosequencing to analyze the DNA methylation levels of each CpG site in 174 the core promoter region.
The reader may ask whether you excluded the core promoter region in subchapter 2.4.
Please clarify this difference.
You should make clear which CpG sites were pyrosequenced in subchapter 2.4. Here you mention CpG sites 1-5 and in 2.5, CpG sites 1-8.
Also indicate the core promotor region in subchapter 2.5.
Please make clear the nomenclature of the CpG sites. A figure could help here to show the different CpGs. Please use a nomenclature that the reader can see where the respective CpG sites are located in the sheep genome (assembly should be referenced) and which are pyrosequenced.
Line 446: Our studies revealed the effects of photoperiod on the GNAQ gene expression and ovine reproductive activity from the perspective of epigenetics,
Conclusions may be improved as the hypothesis stated relies on previous reports. Other hypotheses or influencing factors which may contradict the hypothesis stated should be discussed and based on this discussion, possible limitations of the present study should be outlined.
Comments on the Quality of English LanguageNo comments
Author Response
Comments
- Propose that authors provide some more data on genes influencing season in sheep and the HPG axis. This way the reader can get a better impression on the potential role of GNAQ. A supplementary table would be useful.
Response: Thank you for your suggestion, we will follow up with more progress, so we will not provide this information for now.
- Lines 136-145: please add fragment length for pSI-P1 to pSI-P5.
Response: Thank you for your suggestion, we will follow up with more progress, so we will not provide this information for now.
- Line 152: We verified the DNA methylation level of each CpG site of GNAQ by pyrosequencing, and the results are shown in Figure 4. And in Line 174: We used pyrosequencing to analyze the DNA methylation levels of each CpG site in 174 the core promoter region.The reader may ask whether you excluded the core promoter region in subchapter 2.4. Please clarify this difference.
Response: Thanks for your suggestion, we have already validated the core promoter region in subchapter 2. 3, so we did not repeat the mention in subchapter 2.5.
- You should make clear which CpG sites were pyrosequenced in subchapter 2.4. Here you mention CpG sites 1-5 and in 2.5, CpG sites 1-8. Also indicate the core promotor region in subchapter 2.5.
Response: Thank you for your suggestion, we have made the appropriate changes.
- Please make clear the nomenclature of the CpG sites. A figure could help here to show the different CpGs. Please use a nomenclature that the reader can see where the respective CpG sites are located in the sheep genome (assembly should be referenced) and which are pyrosequenced.
Response: Thank you for your suggestion, we renamed the CpG site for the sake of clearer exposition.
- Line 446: Our studies revealed the effects of photoperiod on the GNAQ gene expression and ovine reproductive activity from the perspective of epigenetics. Conclusions may be improved as the hypothesis stated relies on previous reports. Other hypotheses or influencing factors which may contradict the hypothesis stated should be discussed and based on this discussion, possible limitations of the present study should be outlined.
Response: Thank you for your suggestion, we have already elaborated on it in the Discussion section, so we wrote the conclusion in Conclusions.
